# Telomeres and Age-Related Diseases

**DOI:** 10.3390/biomedicines9101335

**Published:** 2021-09-27

**Authors:** Hans-Jürgen Gruber, Maria Donatella Semeraro, Wilfried Renner, Markus Herrmann

**Affiliations:** Clinical Institute of Medical and Chemical Laboratory Diagnostics, Medical University of Graz, 8036 Graz, Austria; hans.gruber@medunigraz.at (H.-J.G.); maria.semeraro@medunigraz.at (M.D.S.); markus.herrmann@medunigraz.at (M.H.)

**Keywords:** telomere, age, disease

## Abstract

Telomeres are at the non-coding ends of linear chromosomes. Through a complex 3-dimensional structure, they protect the coding DNA and ensure appropriate separation of chromosomes. Aging is characterized by a progressive shortening of telomeres, which compromises their structure and function. Because of their protective function for genomic DNA, telomeres appear to play an important role in the development and progression of many age-related diseases, such as cardiovascular disease (CVD), malignancies, dementia, and osteoporosis. Despite substantial evidence that links telomere length with these conditions, the nature of these observations remains insufficiently understood. Therefore, future studies should address the question of causality. Furthermore, analytical methods should be further improved with the aim to provide informative and comparable results. This review summarize the actual knowledge of telomere biology and the possible implications of telomere dysfunction for the development and progression of age-related diseases. Furthermore, we provide an overview of analytical techniques for the measurement of telomere length and telomerase activity.

## 1. Introduction

Telomeres, the ends of linear chromosomes, have already been studied for over half a century, and today we possess detailed knowledge of the structural organization and physiology [1]. One key feature of telomeres is that they shorten with advancing age, which compromises their structure and function. Over the last decade, numerous studies have reported associations between telomere length and a broad range of age-related diseases including cardiovascular disease, malignancies, dementia, osteoporosis, and others [2]. However, the nature of these relationships and potential molecular mechanisms that may explain them are still insufficiently understood. A particular area of interest in this context is cancer, where genomic stability, cell differentiation, and proliferation is often compromised. Because of their protective function for genomic DNA, telomeres appear to play an important role in the development and progression of malignancies. This review summarize the actual knowledge of telomere biology and the possible implications of telomere dysfunction for the development and progression of age-related diseases.

## 2. Telomeres—Structure and Functions

Telomeres are DNA regions of variable length at the end of all chromosomes. In humans, they are composed by numerous repeats of the hexanucleotide TTAGGG and are organized in a complex 3-dimensional structure, which is essential for the protective properties of telomeres. While telomeres are double-stranded for most of their length, the very end of the leading strand is single-stranded. This single stranded overhang is the result of an incomplete lagging strand DNA synthesis, which leads to telomere shortening with every cell division, beyond telomere shortening due to accidental damage. When telomeres become critically short, cells no longer divide. In 1961, Hayflick discovered that the number of cell divisions in vitro is limited, known as the Hayflick limit [3,4]. In order to prevent inappropriate recognition of the single-stranded overhang as DNA damage and subsequent activation of the DNA damage repair (DDR) system, it is hidden inside the 3-dimensional telomere structure. Inappropriate activation of the DDR system at telomeric sites would results in non-homologous end-joining, alternative non-homologous end-joining or homologous recombination [5]. Because of the progressive shortening of telomeres due to aging, they are often referred to as a “molecular clock of aging.”

Telomere function and maintenance is tightly linked to the shelterin protein complex, which consists of six individual proteins [6]. A detailed description of the shelterin proteins is given in Table 1. This nucleo-protein complex is attached to telomeres and forces double-stranded telomeric DNA to fold back, forming the so called T-loop. Furthermore, with the help of shelterin proteins, the single-stranded DNA overhang at the end of the T-Loop is hidden inside the D-loop, a short section where double-stranded telomeric DNA drifts apart (Figure 1). When telomeres become critically short, formation of the protective T-loop structure is no longer possible, which would expose the single-stranded overhang to the DDR system. To prevent the adverse consequences, like destruction of the genome a DDR signal appears, cells arrest their proliferation cycle and gradually go into senescence.

The compact DNA-structure of telomeres represses the expression of nearby genes through spatial hindering. This transcriptional silencing is known as telomere position effect (TPE) [11,12]. Telomeric motifs are also interspersed between genes of the coding DNA. These interstitial telomeric sequences (ITS) can interact with telomere-associated shelterin proteins, especially telomeric repeat binding factor 2 (TERF2), resulting in the formation of interstitial telomeric loops (ITL) [13]. These ITLs contribute to the complex 3-dimensional chromatin structure and permit telomeres to modify the expression of subtelomeric and distal genes. The latter is referred to as telomere position effect over long distances (TPE-OLD) [14].

## 3. Telomerase

Telomerase is a ribonucleoprotein complex that is able to elongate telomeres through the de-novo synthesis of telomeric DNA and thereby counteracting the end-replication problem [15]. It consists of the protein component telomerase reverse transcriptase (TERT), which harbors the enzyme activity, and the telomerase RNA component (TERC), also referred as human telomerase RNA (hTR). TERC serves as the template for telomere elongation. In early embryonic development, telomerase is active and ensures appropriate telomere elongation. However, within 18 weeks of gestation, the enzyme becomes inactive. In contrast, single-cell eukaryotes require a constantly active telomerase to enable continuous cell division. Telomerase silencing during embryonic life is believed to be mediated either by alternative splicing or epigenetic modifications that alter the 3-dimensional chromatin structure [16,17]. Telomerase activity in humans is primarily regulated through the expression TERT [18]. The regulation of TERC transcription is largely unknown. TERC belongs to the family of non-coding small Cajal body RNAs (scaRNA) and small nucleolar RNAs (snoRNA) and has its own promotor, which is in contrast to most scaRNAs and snoRNAs [19,20,21]. Assembling of functional telomerase depends on the structure of TERC, which can allow or prevent TERT binding [22,23]. In particular, TERT is characterized by 3 domains, the N-terminal domain, which is the telomerase RNA binding domain, the C-terminal domain, and the reverse transcriptase domain. These domains build a ring structure allowing RNA-DNA hybridisation and DNA synthesis. Through the action of additional proteins, the holoenzyme is recruited to telomeres and becomes fully activated [24,25,26]. Firstly, two dyskerin complexes bind to TERC followed by nucleolar protein 10 (NOP10) and NHP2. Finally, assembling of the holoenzyme is completed through association of this protein complex with WD repeat containing antisense to TP53, which regulates catalytic activity. Beyond these co-factors, several other proteins are at least transiently involved for biogenesis and assembling of the telomerase holoenzyme, including SHQ1, pontin, reptin, ATPases, the chaperones HSP90, and TriC [27,28,29,30]. Two recent publications on the structure of substrate bound telomerase using cryo electron microscopy give essentially new insights by revealing a bilobal structure [23,31]. The bilobal structure is scaffold by the RNA component and consists of the catalytic core with TERT and DNA-bound TERC as well as the H/ACA lobe. This lobe is essentially involved in the assembling and contains an H/ACA protein complex with dyskerin, NOP10, GAR1, NHP2, and telomerase Cajal body protein 1 (TCBA1).

As described above, telomeres are associated with the shelterin complex to protect the telomeric DNA from inappropriate DNA repair response. Moreover, the shelterin complex is also involved in the recruitment of telomerase to the telomeric DNA. This recruitment is cell cycle dependent and occurs in the S phase [32] through binding of TERT to the shelterin protein tripeptidyl peptidase 1 [33,34,35,36]. TEN and TEL are the specific protein domains that mediated the binding of TERT to TPP-1 [37].

Taken together, the telomeric region represents a highly complex DNA-protein structure, consisting of telomeric DNA and the shelterin complex. Through interaction of the shelterin proteins with telomeric DNA, a complex 3D structure is formed that protects telomeres from inappropriate DDR activity and regulates the expression of subtelomeric and distal genes. Telomeres shorten naturally with every cell division due to the end-replication problem and accidental damage. The telomerase enzyme complex can counteract telomere shortening through de-novo synthesis of telomeric DNA.

## 4. Influencing Factors of Telomere Length

In humans, mean leukocyte telomere length (LTL) at birth is 11 kilo base pairs (kbp) and declines to less than 4 kbp in elderly individuals [2]. However, telomere shortening is not a linear process, where a constant number of base pairs is lost with every cell division. Telomerase activity and telomere trimming events can modulate telomere length in both directions. Numerous studies have investigated non-modifiable and modifiable factors that influence telomere length. Gender, for example, is a non-modifiable factor that determines telomere length with longer telomeres being observed in females than in males [38]. This effect is mainly driven by estrogen, which mediates antioxidative effects and induces moderate telomerase activity. Psychological stress is another well-documented factor that impacts telomere homeostasis by reducing telomerase activity and increasing reactive oxidative species [39,40]. Also, nutritional factors can modulate telomere length [41,42,43]. In particular, a sufficient supply with micronutrients like vitamin A, D, C, E, B12, folate, and nicotinamide is positively associated with telomere length [44,45,46,47,48]. Minerals like magnesium, zinc, and iron, and other dietary components, such as omega-3 fatty acid, polyphenols, and curcumin, are additional modulators of telomere length. The effects of vitamins on telomere homeostasis seem to be mediated by their antioxidative properties and the prevention of DNA damage. In addition to a healthy diet, regular physical activity also contributes to the preservation of telomere length via reducing sustained oxidative stress and inflammatory mechanisms. Furthermore, exercise has been shown to increase telomerase activity [49,50,51]. Other lifestyle-related factors that potentially influence telomere length include smoking and alcohol consumption. However, to date the evidence for a significant associations between alcohol consumption and telomere length is insufficient [52]. Regarding smoking, a recent meta-analysis of 84 studies showed significantly shorter telomeres in ever smokers compared to those who never smoked [53]. Taken together, there is good evidence that a healthy and active lifestyle with sufficient sleep and a low psychologic stress level contributes to the preservation of telomeres. Physical inactivity, nutritional deficits, overweight, stress, and smoking can accelerate telomere shortening and thus promote age-related diseases.

## 5. Telomeres and Age-Related Diseases

Aging is characterized by progressive telomere shortening due to cell division and telomere erosion. Individuals of the same age with the shortest telomeres have compared to those with the longest telomeres a higher hazard ratio for all-cause mortality [54,55]. Furthermore, telomere length is also related to the incidence, progression, and disease-specific mortality of individual age-related diseases, such as CVD, type 2 diabetes, cancer, and Alzheimer’s disease [56]. These associations are believed to be the result of the age-related telomere shortening, which contributes to genomic instability and modulates gene expression through TPE, TPE-OLD, and DDR activation [57]. Critically short telomeres cannot loop back to form genomic ITS, which silence nearby genes. As a result, the expression of these genes is increased [58,59,60]. Interestingly, one of the genes that is regulated via TPE-OLD and ITL is TERT, which encodes the telomere elongating enzyme telomerase [61]. The activation of TERT expression in the context of short telomeres is considered as a protective mechanism that prevents rapid telomere shortening. It is believed that the physiologic stimulation of telomerase activity through physical activity, healthy nutrition, and other modifiable lifestyle factors can reduce the risk for age-related diseases and promote healthy aging [51]. This concept is supported by experimental studies in mice showing that constitutive TERT expression delays aging and extends life span [62,63,64]. However, it is also well established that the constitutive expression of TERT is strongly associated with carcinogenesis and TERT inhibition in cancer cells reduces tumour growth due to the induction of cell death [65,66,67,68,69]. This suggests that only the physiologic stimulation of TERT may have beneficial effects [62,63,64]. However, this assertion is discussed controversial as activation of telomerase as therapeutic target may have beneficial effects in telomere-shortening-associated conditions [70].

Another mechanism involved in the shortening of telomeres is the clonal hematopoiesis of undetermined potential (CHIP) [71]. Naturally occurring somatic mutations in hematopoietic stem cells accumulate with advancing age and can induce the clonal expansion of mutated leucocytes. The prevalence of CHIP is very low in subjects younger than 40 years, but increases up to 10% and 20% by the age of 70 and 80 years, respectively [72]. Individuals with CHIP have significantly shorter telomeres, and moreover, a whole-genome analysis study including 11,262 participants revealed that CHIP showed the strongest association with the TERT gene, in particular a germline deletion in intron 3 [73]. Clinically, CHIP carriers have a higher risk for haematological malignancy and adverse cardiovascular events [74,75,76].

## 6. Cardiovascular Diseases (CVD)

In developed countries, CVD is one of the most frequent age-related diseases and represents the leading cause of death. The prevalence of cardiovascular diseases (CVD) and the frequency of CVD-related complications increases with advancing age. Expression of TERT and the presence of telomerase activity in cardiomyocytes and blood vessels have nurtured the idea that telomeres might be of particular importance for cardiovascular aging [77,78,79]. Over the last two decades, numerous human and animal studies have investigated the role of telomeres and telomerase in CVD with inconsistent results. Most of them measured telomere length in leucocytes (LTL) as a surrogate marker for telomere length in solid tissues, which are not easily accessible. Several prospective cohort studies have found an elevated risk of CVD, myocardial infarction, heart failure, and stroke in individuals with a low LTL and a high telomere attrition rate [80,81,82,83,84,85]. Amongst 800 men and women who participated in the prospective, population-based Bruneck study, LTL was an independent predictor of myocardial infarction and stroke [86]. CVD risk differed between individuals with the longest and shortest LTL by a factor of 2.72. However, with only 88 CVD events, the number of end-points was rather low. Consistent with the Bruneck study, in over 1500 Scottish people with and without CVD, individuals in the lowest and the middle tertile of LTL had a 40–50% higher risk for incident CVD during follow-up [80]. Another study of patients with and without early myocardial infarction (<50 years) found that LTL of patients was equivalent to controls that were 11.3 years older [87]. An interesting study by Benetos et al. showed that the association of short telomere length and atherosclerotic cardiovascular disease is based on a higher telomere attrition rate in early life [88]. This observation alludes to additional telomere-related effects in CVD that go beyond normal age-related telomere shortening. In addition to coronary atherosclerosis and myocardial ischemia, the risk of ischemic, atherothrombotic, and haemorrhagic stroke also seems to be associated with telomere length [89,90,91,92,93]. Also, Martin-Ruiz et al. found that telomere length is associated with post-stroke mortality [94]. However, not all studies corroborate an association of LTL with CVD and stroke [95,96,97,98,99]. To address the inconsistencies of existing studies, Haycock et al. performed a meta-analysis of 24 studies with over 43,000 participants and 8400 CVD patients [100]. Individuals with the shortest LTL had a 54% higher CVD risk than those with the longest LTL, and when considering only prospective studies relative risk of CVD was still 40% higher in individuals with short telomeres. Despite their significant results, meta-analyses should be interpreted with caution as the studies included varied substantially in study design and patient characteristics. This is particularly important when considering that CVD is a highly multifactorial entity that is associated with multiple risk factors, such as age, gender, obesity, physical inactivity, genetic predisposition, and others, and involves several pathomechanisms including inflammation, oxidative stress, and dyslipidemia [100,101].

Telomere length and telomerase activity show huge interindividual variability, and as a result, most human cohort studies found only a weak correlation between LTL and age. Considering the many factors that affect telomere length, this observation is not surprising. As discussed earlier, LTL has been reported to be influenced by many lifestyle factors including sleep, physical activity, psychological stress, and nutritional factors [48,102,103,104]. Furthermore, oxidative stress and chronic inflammation are key factors that impact telomere length [47]. Besides many lifestyle and environmental factors, telomere length and telomerase activity are strongly determined by the inherited genetic background, with heritability estimates ranging from 34% to 82% [105]. Interestingly, the maternal influence on telomere length seems to be stronger than the paternal influence. This genetic basis can be used for Mendelian randomization (MR) studies to evaluate the potential causal relationship between telomere length and age-relates diseases. In an analogy to randomized clinical trials, MR creates study groups stratified by genotypes, which are independent of confounding factors and are inherited at random. MR studies are more robust to confounding factors or reverse causation than observational studies [106]. In a large MR study including more than 261.000 participants, Kuo and coworkers could show a modest causal association between LTL and lower CVD and cancer risk. However, no causal association of LTL with other age-related health outcomes was found [107]. Furthermore, GWAS studies have identified seven SNPs that are responsible for interindividual variations in LTL. The presence of these alleles seems to be related with an increased CVD risk. In a meta-analysis by Codd et al. one standard deviation reduction in LTL was associated with a 21% higher CVD risk [108].

Chronic inflammation and oxidative stress are critical factors that promote atherosclerosis. In addition, they accelerate telomere shortening and cause premature cellular senescence in endothelial cells, vascular smooth muscle cells (VSMC) and blood leukocytes [109]. The severity of CVD seems to be correlated with the reduction of telomere length in VSMCs of human atherosclerotic plaques. Also, VSMCs of atherosclerotic plaques are characterized by oxidative DNA damage and the expression of typical senescence markers, such as senescence-associated galactosidase, cyclin-dependent kinase inhibitors p16 and p21, decreased expression of cyclin D and cyclin E, and hypophosphorylation of the retinoblastoma protein [110]. Senescent VSMCs harbor a limited proliferative capacity and an increased amount of matrix-degrading enzymes, which stimulate the thinning of fibrous caps and plaque rupture [111]. In patients with acute coronary syndrome, a low LTL is linked to the presence of highly unstable atherosclerotic plaques and an increased proinflammatory activity [112]. Accelerated telomere shortening also goes along with functional exhaustion and impaired proliferative capacity of bone marrow-derived endothelial progenitor cells. These cells are of critical importance for the re-endothelialization process of damaged blood vessels. As a result, re-endothelialization after vascular injury and stent implantation is delayed in individuals with short telomeres [113]. Another piece of evidence that underpins the role of telomeres in CVD is the observation that statins, which are commonly used to treat hyperlipidaemia, modify TERT expression and telomerase activity in the vascular wall and in cells of the immune system [114]. Furthermore, in an experimental mouse model, long telomeres were protective against age-dependent cardiac disease caused by NOTCH1 haploinsufficiency [115].

Beyond telomeres, telomerase is also discussed to be involved in cardiovascular diseases due to its noncanonical and nonnuclear functions. Telomerase is also present in mitochondria, thereby improving membrane potential, reducing mitochondrial reactive oxygen species (ROS) production. These effects counteract the induction of apoptosis by protecting mitochondrial DNA [116,117,118]. In preclinical models, telomerase has also been shown to be involved in autophagy through inhibition of mammalian target of rapamycin complex 1 (mTORC1), especially under calorie restriction conditions, thereby improving diastolic dysfunction [119,120].

In summary, there is substantial evidence that links short telomeres and accelerated telomere shortening to CVD. However, at present it is still unclear whether telomeres are causally involved in the development and progression of CVD or if this association simply represents an epiphenomenon.

## 7. Type 2 Diabetes (T2DM)

T2DM, a common risk factors for CVD, is a multifactorial disease that is mainly driven by obesity and physical inactivity. Several studies have shown that individuals with short telomeres have a higher risk of T2DM, a faster disease progression and more diabetic complications, such as retinopathy, nephropathy, neuropathy, and peripheral vascular disease [121,122,123,124,125,126,127,128,129]. Also, a recent meta-analysis by D´Mello et al. supports a significant association between telomere length and T2DM [130]. Short telomeres in T2DM seem to go along with epigenetic modifications. In Chinese diabetics, short telomeres have been found to be associated hyper-methylation of LINE-1, a surrogate marker for global DNA methylation [131]. Furthermore, in TERT knockout mice with premature telomere shortening, the proliferative senescence of adipose progenitor cells contributes to ageing, obesity, and diabetes [132]. An autopsy study by Tamura et al. revealed shorter telomeres in the beta-cells of T2DM patients than in non-diabetic individuals [133]. While the studies discussed before support an association between telomeres and T2DM, others do not. In a recent general population study with 3921 participants, Menke et al. found no association of LTL with diabetes status, diabetes duration, or diabetes control [134]. Likewise, in the Finnish Diabetes Prevention Study, Hovatte et al. found no association between short telomeres and the development of T2DM during 8.5 years of follow up [135]. A MR study by You et al. also failed to show an association between telomere length and T2DM risk [136]. However, a direct comparison of different studies is limited by the substantial heterogeneity of their study design and the age, ethnicity, sex, and health status of the participants.

In summary, there is some evidence indicating an association of accelerated telomere shortening with T2DM. However, existing studies are inconsistent and mainly of observational nature. Therefore, the role of telomere function in T2DM remains unclear and more research is needed to understand if there is a mechanistic link.

## 8. Cancer

Similar to CVD and T2DM, the incidence of cancer also increases with age. Considering the protective role of telomeres for genomic DNA, numerous studies have investigated if telomere length is related to cancer risk or prognosis [2,137]. Existing results suggest a dual role of telomeres in neoplastic diseases. Short telomeres seem to increase the risk for incident cancer. On the other hand, with the malignant transformation it is believed that critically short telomeres contribute to cancer progression and the reactivation of telomerase [138]. In contrast to somatic cells, up to 90% of human tumor tissue is characterized by telomerase activity [18]. This telomerase reactivation is a tumor escape mechanism that confers immortalization to affected cells and promotes tumor invasion and metastasis.

There is solid evidence that the telomeres from cells of cancerous lesion are shorter than in healthy tissue from the same organ [139,140,141,142]. Furthermore, telomeres from peritumoral tissue are shorter than from more distant areas [143]. It has also been reported that in breast and prostate cancer tissue, short telomeres are related to an advanced disease state at diagnosis, faster disease progression, and poorer survival [139,144]. Most studies ignored the fact that telomere length varies between chromosomes and cells. However, the variability of telomere length seems to be a risk factor for cancer-related death. The practical limitations of telomere length measurement in tumor tissue lead researchers to analyze LTL in most large human studies. In population-based, prospective studies, a low LTL at baseline was associated with a substantially higher risk for incident tumors and tumor-specific mortality [145,146]. Serial LTL measurements, such as in the normative Aging Study, showed that subjects with incident cancer had a markedly higher telomere shortening rate than subjects without [138]. However, there are also studies that did not find significant associations between LTL and cancer risk [147], and others reported a higher LTL in cancer patients than in cancer-free subjects [148]. These inconsistent results are further supported by two meta-analyses of retrospective [149] and prospective studies [150]. While the pooled analyses of retrospective studies showed an inverse relationship between LTL and cancer risk, this was not the case when considering only prospective studies.

Under physiologic circumstances, the natural process of progressive telomere shortening with ongoing cell division and aging ends with critically short telomeres, which induce replicative senescence [151,152]. In this situation, chromosomal ends are no longer protected and therefore recognized by the DDR machinery. Finally, telomere crisis facilitates autophagy and ultimately leads to cell death. Tumor cells can escape this mechanisms by the reactivation of telomerase [153,154,155]. The reactivation of telomerase leads to telomere elongation and thus prevents telomere crisis with subsequent cell death. Various mechanisms for the reactivation of telomerase in tumor cells have been proposed. For example, mutations in the TERT promotor region can increase TERT transcription [156]. Such mutations have been described in various tumour types of tissues with low of self-renewal rates, e.g., melanoma, hepatocellular carcinoma, glioblastoma, and urothelial cancer [157,158,159]. The wildtype promotor of TERT is silenced by methylated histon H3, whereas the mutated allele is associated with non-silencing H3 variants, thus enabling TERT expression [160]. Moreover, transcription is driven by recruitment of the transcription factor GA binding protein transcription factor subunit alpha (GABPA) and subunit beta (GABPB) to the mutated promotor, leading to an increased transcription of the mutated TERT allele in cancer cells [161]. However, the majority of tumors harbor increased telomerase activity without activating mutations. Several other mechanisms, such as gene amplification of TERT, where the number of copies of the TERT gene is increased due to errors in the DNA replication and repair machinery, have been described. For example, TERT gene amplification has been found in ovarian and lung cancer [162]. Chromosome rearrangements can also upregulate TERT expression, such as in neuroblastoma [163,164]. Another intriguing mechanism that can activate TERT expression has been described in hepatocellular carcinoma, where a hepatitis B infection can result in integration of the virus genome near the TERT promotor [165]. In this scenario, TERT expression is increased through viral enhancer elements [165]. Tumour cells can overcome telomere shortening also by alternative telomere lengthening via homologous recombination mechanisms resulting in heterogeneous telomeres. This type of telomere length preservation has been described for neuroendocrine tumours and sarcomas [166]. TPEs, where critically short telomeres induce the expression of genes near the telomeric region may also mediate telomere maintenance in cancer cells [167].

The cancer associated activation of telomerase is of particular interest as it may represent a potential target for novel therapeutic strategies in cancer patients. Several therapeutic approaches aim to antagonize the continuous activation of telomerase in cancer cells [168]. An advantage of telomerase-targeting therapies is the fact that proliferating cancer cells have shorter telomeres compared to normal somatic cells, which should make them more sensitive to anti-telomerase therapeutics [169]. One of the most promising candidates for telomerase-targeted cancer therapy may be the first-in-class telomerase inhibitor Imetelstat, which is currently tested in a phase 3 clinical trial (ImpactMF; NCT04576156) with an estimated enrollment of 320 patients with refractory myelofibrosis (MF). Results of the trial are expected for May 2024.

In summary, telomere biology in cancer is highly complex, and the role of telomeres in the development and perpetuation of cancerous lesions is still subject to intensive research. So far, it seems that short telomeres increase the risk for cancer formation, whereas telomerase reactivation and preserved telomeres are important for tumour growth and survival.

## 9. Alzheimer’s Disease (AD)

In the aging populations of developed countries, dementia and AD affect a rapidly rising number of individuals. Despite a better understanding of the underlying pathological mechanisms, effective therapies for this devastating disease are still lacking. Therefore, a lot of research is focused on risk prediction, early diagnosis, and the identification of modifiable risk factors. Age-related telomere shortening has been proposed to contribute to neuronal dysfunction and cognitive decline in elderly individuals. In this context, previous studies reported shorter LTL in AD patients than in non-demented controls [170]. According to Panossian et al., the inverse relationship between LTL and AD risk is primarily driven by T lymphocytes, but not B lymphocytes or monocytes [171]. Furthermore, telomere length in T cells is inversely correlated with serum levels of TNF-α, with the proportion of CD8+ T cells lacking expression of the CD28 co-stimulatory molecule, and with apoptosis. In a recent meta-analysis, Forero et al. confirmed a significant difference in LTL between AD patients and controls [172]. However, most of the studies considered in this meta-analysis were of cross-sectional nature and included a rather small number of AD patients. Until today, only a few large prospective observation studies have been published Honig et al. analysed LTL in 1,978 elderly individuals from the prospective Washington Heights-Inwood Community Aging Project, and found significantly shorter LTL in participants with prevalent or incident dementia [173]. In 1,961 elderly participants of the Rotterdam study, LTL showed a U-shaped association with AD and all-cause dementia [174]. While the increase of AD risk in individuals with short telomeres is in line with previous studies, the link between over-elongated telomeres and AD risk is not yet understood and needs further clarification. Experimental studies that may shed some light into potential mechanisms are largely lacking. In one of the few existing studies, neurons of TERT knockout mice showed shorter telomeres, an increased production of oxidative species and an increase in cellular oxidative damage in response to tau [175]. A higher AD risk in individuals with long telomeres is supported by the observation that over-elongated telomeres in human embryonic stem cells become fragile and accumulate DNA damage [176]. According to this study, telomere homeostasis is not only regulated by telomerase activity, but also by telomere trimming events.

Tedone et al. reported that LTL is not only associated with AD risk, but also with disease progression [177]. Late onset AD patients with slow disease progression had shorter telomeres than those with fast disease progression or healthy elderly controls. Although there is substantial evidence for an association between LTL and dementia, an analysis of telomere length in matched pairs of peripheral blood leucocytes and cerebellum samples from AD patients and unaffected controls questions a causal relationship [178]. While LTL and telomere length in the cerebellum of AD patients were correlated, the latter did not differ between AD patients and controls.

Taken together, existing studies provide robust evidence for an association between LTL and AD risk. However, the nature of this relationship is largely unclear. It remains to be elucidated whether low LTL in AD patients represents a cumulative marker of various AD risk factors or if short telomeres are mechanistically involved in the development and progression of dementia.

## 10. Osteoporosis

Like many other tissues, also bone cells exhibit an age-related decline in telomere length. In addition, premature aging syndromes, such as Werner syndrome and congenital dyskeratosis, are characterized by telomere dysfunction and osteoporosis. These observations have led to the hypothesis that the age-related shortening of telomeres does also contribute to the development and progression of osteoporosis, a highly prevalent condition amongst elderly individuals [179]. A large population-based study in 2150 women between 18 and 79 years supports this idea by showing a significant correlation between LTL and bone mineral density (BMD) at the spine and forearm [180]. Furthermore, risk of clinical osteoporosis was lower in women with longer telomeres. The difference in LTL between women with and without osteoporosis was equivalent to five years of telomeric aging. Also in elderly men (71–86 years), LTL correlated with bone loss at the distal forearm [181]. In vitro experiments with cultured mesenchymal stroma cells showed a reduced proliferative and osteogenic capacity in osteoporotic patients [182]. Additional support for a causal link between osteoporosis and short telomeres comes from an in vivo study in telomerase deficient mice [183]. The accelerated telomere shortening in these animals was associated with bone loss due to dysfunctional osteoblasts and osteoclasts. In another mouse study of accelerated aging, Wang et al. showed an impairment of osteoblast differentiation due to proliferation-independent telomere dysfunction [184].

The studies discussed before are contrasted by several other studies that did not find significant relationships between LTL, BMD and osteoporosis [185,186,187]. Two large-scale observational cohort studies in community-dwelling women failed to show significant associations of BMD, bone loss, and prevalent and incident fractures with LTL [186,187]. Also, a comparison of telomere length in human osteoblasts and peripheral blood leucocytes from young, elderly, and osteoporotic women did not support the concept of accelerated telomere shortening and premature cellular aging in osteoporotic patients [188]. In a recent review, Fragkiadaki et al. stated that despite a number of promising studies there is still a lot of inconsistencies in the available literature [179]. Therefore, without further observational and experimental studies the impact of age-related telomere shortening on bone aging and osteoporosis cannot be established.

## 11. Analytical Aspects

The methods for TL measurements vary widely as traditional techniques have been refined and new techniques have emerged [189]. Beyond the terminal restriction fragment (TRF) analysis by Southern blot as the gold standard, quantitative fluorescence in situ hybridization with flow cytomerty (qFISH) and PCR applications such as qPCR, STELA (Single Telomere Length Assay) and TESLA (Telomere Shortest Length Assay) are used for TL analyses [190,191,192,193]. qPCR is the most commonly used method for telomere length analysis, because it is cheap and can easily be performed in a large number of samples. As the result is the ratio of relative TL to a single copy gene, this method gives no information about the shortest telomeres or differences between chromosomes. In contrast, qFISH analysis in combination with TRF determines absolute telomere length and provides information on the shortest telomeres. This is of particular interest as it is hypothesised that not average telomere length, but the shortest telomeres drive chromosomal instability and reduce cell viability [194]. This method seems to have the potential to provide more disease-related information on alterations of telomere length. However, the method is laborious and therefore hard to use for large sample analysis. For analysis of telomerase activity the telomeric repeat amplification protocol (TRAP) is most commonly used. It consists in a two-step PCR including telomerase mediated primer amplification and subsequent qPCR analysis of the amplification products. To analyze TA in single cells a droplet digital TRAP (ddTRAP) assay was developed by Ludlow et al. [195], which provides more sensitivity and better throughput [196]. However, TA assays are hard to standardize and not practicable for clinical studies. Most clinical studies are also limited by the fact that the source for telomere analyses are whole blood samples, and therefore, TL and TA of blood leucocytes is determined [197,198,199]. Leucocytes represent a heterogeneous cell population with a highly variable composition, depending on environmental and lifestyle factors like nutrition, physical activity, and psychological stress. To date, it is not clear if leucocytes are a suitable surrogate to investigate telomere dynamics in other solid organs. A recent analysis of 6391 samples from the Genotype-Tissue Expression (GTEx) project showed that LTL correlates with the telomere length of some, but not all, somatic tissue types [200].

## 12. Conclusions

There is a large body of evidence that supports significant associations between telomere length and age-related diseases, such as CVD, T2DM, cancer, dementia, and osteoporosis. However, existing studies are heterogeneous, and a substantial number of studies did not find significant relationships. Furthermore, most clinical studies are limited by the measurement of average telomere length in peripheral blood leucocytes, which may not necessarily reflect the situation in other solid organ tissues, such as myocardium, blood vessels, brain, or bone. Mechanistic studies that confirm a mechanistic involvement of short telomeres in common age-related diseases are largely lacking. Last but not least, available methods for the measurement are poorly standardized and provide different information, which strongly limits the comparability between studies. Future research projects should focus on the question of causality. Also, methods should be selected on the basis of their analytical capabilities and harmonized wherever possible.

## Figures and Tables

**Figure 1 biomedicines-09-01335-f001:**
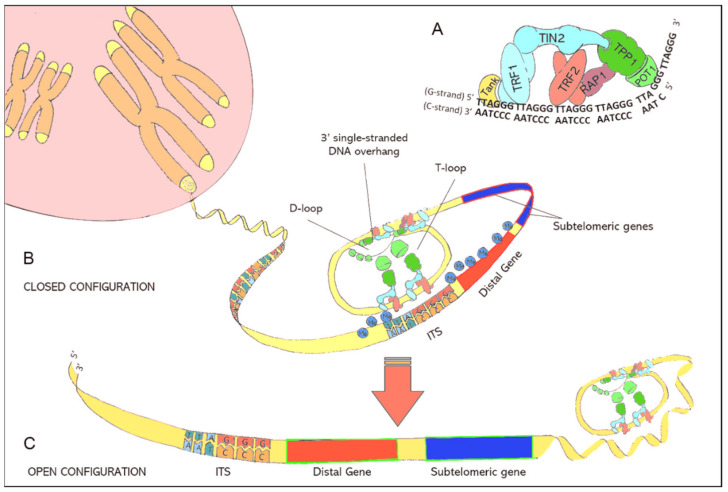
(**A**): Illustration of the shelterin nucleoprotein complex, which protects coding DNA and discriminates telomeres from other free DNA ends resulting from DNA damage. (**B**): Illustration of the closed chromatin configuration at telomeric ends. Due to protein-protein interactions and the specific binding of shelterins to telomeric DNA, double-stranded telomeric DNA is forced to fold back in a loop structure (T-loop) while the 3′ single-stranded DNA overhang is hidden inside the D-loop. The binding of shelterins to interstitial telomeric sequences (ITS) leads to the formation of interstitial telomeric loops (ITL) and establishes a closed chromatin structure that impedes the expression of subtelomeric and distal genes through telomere position effects (TPE). (**C**): lllustration of the open chromatin configuration at telomeric ends. Critically short telomeres can no longer maintain the compact chromatin structure. The resulting open chromatin structure facilitates the access of the translational machinery to genes that were formerly silenced by TPEs.

**Table 1 biomedicines-09-01335-t001:** Components of the Shelterin complex and their functions. [6,7,8,9,10].

Shelterin Protein	Function
Telomeric repeat binding factor 1 (TRF-1)	TRF-1 binds the canonical 5′-TTAGGG-3′ double-stranded telomeric repeats and is important to determine the structure of telomeric ends, as it is implicated in the generation of t-loops and the regulation of telomeric DNA synthesis by the reverse-transcriptase telomerase.
Telomeric repeat binding factor 2 (TRF-2)	TRF-2 is a paralog of TRF-1. As its paralog, TRF-2 has an essential role in maintaining the conformational status of telomeres. It is implicated in telomeric ends protection and telomere length homeostasis.
TRF-1 interacting nuclear protein 2 (TIN-2)	TIN-2 can bridge TRF-1 to the TRF-2/RAP-1 protein complex and recruits the TPP-1/POT-1 heterodimer to telomeric ends. In this way, TIN-2 is important for the assembly of the Shelterin complex and thereby the protection of telomeric ends.
Telomeric overhang binding protein 1 (POT-1)	POT-1 forms with TPP-1 a heterodimeric binding protein, which binds to the single-stranded 5′-TTAGGG-3′ repeats. Thus, it is critically involved in telomere conformational changes. In this way, the interaction between POT-1 and the enzyme telomerase allows the addition of new hexanucleotides to chromosome ends.
TIN-2 and POT-1 interacting protein 1 (TPP-1)	TPP-1 forms a heterodimer with POT-1 and plays an important role in the recruitment of telomerase to telomeric ends.
Repressor-activator protein 1 (RAP-1)	RAP-1 forms a 1:1 complex with TRF-2 and is important for the structure, protection, and elongation of telomeres. As a modulator of the NF-κB signaling pathway, RAP-1 is also involved in the regulation of the energy metabolism.

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
