# Peer review of "Telomeres and Age-Related Diseases"

_biomedicines, 2021, doi:10.3390/biomedicines9101335_

Round 1

Reviewer 1 Report

The manuscript by Gruber et al., is a review paper about the telomere biology and telomere dysfunctions in age-related diseases. This is a very important topic, making any new contribution to the field interesting. Given the complexity involved, the author has produced many positive and welcome outcomes. The literature review offers a useful overview of current research and policy, and the resulting bibliography provides a very useful resource for current practitioners.

Overall, this review is well written, and the content of this manuscript is of major interest. Nevertheless, the following issues need to be addressed:

- The review would be significantly improved with the addition of other illustrations and/or tables. Actually, I believe this journal recommends at least 2 figures and/or tables in the case of review papers.

- Line 8: Maybe you can write “3-dimensional”?

- The references style in the text is wrong. It should be something like [1,2]. Please check the journal guidelines

- Lines 45 and 482: “in vitro” should be in italics

- Lines 94-95: 3-dimensional

- Line 125: Please define here the abbreviation “LTL”

- Please try to be consistent with the references. In lines 292-295 you write: Menke et al (without point and comma), then Hovatte et al and You et al. Then, in line 385 Panossian et al., (with point and comma) and in line 389 Fornero et al. (with point).

- Line 299: This sentence is very similar to that of line 274. Please rephrase it

The reference list is totally wrong. Please check the journal guidelines

Author Response

Dear Reviewer,

thank you for the critical evaluation of our manuscript. According to your suggestions, we have incorporated the following changes in the revised manuscript:

1) We have added a table summarizing components of the shelterin complex and their functions.

2) Line 8: The typo has been corrected.

3) The reference style has been corrected.

4) "in vitro" is now written in Italics.

5) Lines 94-95: The typo has been corrected.

6) Line 126: LTL is now defined as "leukocyte telomere length (LTL)".

7) We have changed references to a consistent style ("author et al.").

8) Line 299: The sentence has been rephrased: "In summary, there is some evidence indicating an association of accelerated telomere shortening with T2DM".

9) The style of the references list has been changed according to the journal guidelines.

Reviewer 2 Report

This work by Gruber et. al. provides a comprehensive review about telomerase and telomeres in age-related diseases. This review is nicely written. I have only few comments:

  1. Since the shelterin complex is shown in figure 1, these proteins should be discussed in more detail due to their vital role in preserving telomeric integrity.
  2. Some of the text in Figure 1 is hard to read. Changing it to a white background will help with the contrast.
  3. Line 132: 'Moderately' should be changed to 'moderate'.
  4. In the cancer section, it would be good to break this down into subsections of specific cancers and discuss telomere specific phenotypes in those specific cancers.
  5. Cryo-EM structures of human and Tetrahymena telomerase are published. Please cite these references and describe key findings in the text: 1) Nguyen et. al. Nature. 557:190–195 (2018) and 2) Jiang et. al. Cell. 173:1179-1190 (2018).

Author Response

Dear Reviewer,

thank you for the critical evaluation of our manuscript. According to your suggestions, we have incorporated the following changes in the revised manuscript:

1) In the revised manuscript we have added a table that provides more detailed information on all shelterin proteins including their function.

2) As suggested by the reviewer, the background color of figure 1 has been changed from black to white for improved legibility of the text.

3) 'Moderately' has been changed to 'moderate'.

4) We have attempted to break down the cancer section into cancer-specific subsections, but the high number of different cancer subtypes resulted in a more confusing structure, and a more profound presentation of telomere-related data for all cancer sub-types would be beyond the scope of our review. We would therefore like to suggest to leave the cancer section without additional cancer-specific subsections.

5) Key findings and references of papers describing cryo-EM structures of telomerase have been added on page 4, lines 113-119:

‘Two recent publications have provided novel insights into the structure of substrate bound telomerase using. [23, 31]. With the help of cryo electron microscopy a bilobal structure has been revelaed. The bilobal structure is scaffold by the RNA component and consists of the catalytic core with TERT and DNA bound TERC as well as the H/ACA lobe. This lobe is essentially involved in the assembling and contains an H/ACA protein complex with dyskerin, NOP10, GAR1, NHP2 and telomerase Cajal body protein 1 (TCBA1).’